# Long-Term Monitoring of Water and Air Quality at an Indoor Pool Facility during Modifications of Water Treatment

Lester T. Lee [1] and Ernest R. Blatchley III [1,2,*]

1   Lyles School of Civil Engineering, Purdue University, West Lafayette, IN 47907, USA; lee1712@purdue.edu
2   Division of Environmental and Ecological Engineering, Purdue University, West Lafayette, IN 47907, USA
*   Correspondence: blatch@purdue.edu

**Abstract:** Previous research has shown that volatile disinfection byproducts (DBPs) can adversely affect the human respiratory system. As a result, swimming pool water treatment processes can play important roles in governing water and air quality. Thus, it was hypothesized that water and air quality in a swimming pool facility can be improved by renewing or enhancing one or more components of water treatment. This study is designed to identify and quantify changes in water and air quality that are associated with changes in water treatment at an indoor chlorinated swimming pool facility. Reductions in aqueous trichloramine ($NCl_3$) concentration were observed following the use of secondary oxidizer with its activator. This inclusion also resulted in significant decreases in the concentrations of cyanogen chloride (CNCl) in pool water. The concentration of urea, a compound that is common in swimming pools and that functions as an important precursor to $NCl_3$ formation, as well as a marker compound for the introduction of contaminants by swimmers, was also reduced after the addition of the activator. Concentrations of gas-phase $NCl_3$ did not decrease after the treatment processes were changed. The collection of long-term water and air quality measurements also allowed for an assessment of the effects of bather load on water and air quality. In general, the concentrations of urea (an $NCl_3$ precursor), liquid-phase $NCl_3$, and gas-phase $NCl_3$ all increased during periods of high swimmer number.

**Keywords:** swimming pool water chemistry; volatile disinfection byproducts; indoor air quality; water treatment process



## 1. Introduction

Swimming is a year-around activity in temperate and cold regions due to the construction of indoor pool facilities. Swimming pool water must undergo treatment to achieve clean and clear water as well as degrade, remove, or inactivate harmful substances, including waterborne microbial pathogens, such as viruses, bacteria, and protozoa [1–3]. Chlorine is the most frequently applied disinfectant and oxidant in swimming pools because of its relatively low cost and ease to use [3–5]. However, chlorine is also known to react with various organic and inorganic compounds to produce disinfection byproducts (DBPs) in swimming pools [3]. Chlorine has been demonstrated to react with several compounds that are attributed to the (poor) hygiene practices of swimmers, leading to the formation of DBPs in chlorinated pools. Specifically, numerous organic-N compounds that are found in body fluids (sweat, urine, and saliva) have been identified as precursors of volatile DBPs that are common in swimming pools. These precursor compounds include urea, creatinine, amino acids, and uric acid [3,6,7]. Several gas-phase DBPs in indoor pool air have been associated with adverse health effects among swimmers and pool workers [4,8–11].

Among volatile DBPs that are common to swimming pools, trichloramine ($NCl_3$) has been examined most extensively due to its association with irritation of the respiratory system in chlorinated swimming pools [12–14]. Additionally, gas-phase $NCl_3$ is largely responsible for the chlorine odor in indoor swimming pool facilities. $NCl_3$ can irritate the

eyes and skin in some people [14–16]. Long-term investigations of gas-phase $NCl_3$ concentration in indoor chlorinated swimming pools have been conducted in recent years [17–20]. However, most of these previous investigations have incorporated analytical methods that were only able to report measurements every 30 min or more [21]. In this study, gas-phase $NCl_3$ is measured with a Next Environmental Monitoring (NEMo) Indoor Air Quality (IAQ) monitor (NEMo XT, Ethera Labs, Crolles, France), which reports measurements of gas-phase $NCl_3$ concentration every 10 min [22,23].

Swimming pool water treatment processes can play significant roles in water and air quality. Moreover, because several important constituents of swimming pool water (including several DBPs) are volatile, there are strong links between water quality and air quality in chlorinated indoor pools. Thus, it was hypothesized that water and air quality in indoor pools can be improved by modifying one or more components of pool water treatment.

In this study, two treatment components are selected to examine this hypothesis: filter media and secondary oxidizers. The experiments were carried out to characterize and quantify the effects of subsequent treatment component changes on water and air quality in an indoor chlorinated swimming pool. Three modes of pool water treatment are examined: the original operation functioned as the baseline of this study (i.e., experimental control) and was monitored for roughly three weeks; the first treatment change involved the replacement of sand filter media to activated filter media, which was monitored for roughly four weeks; the second treatment change involved the introduction of a secondary oxidant and activator, with water and air quality being monitored for a period of eight weeks.

The time frames applied for each stage were selected to allow water and air quality following each process change to approach stable conditions and to examine the effects of these treatment process changes over a range of pool use conditions. For each stage of operation, samples of water and air were collected on a regular schedule. The purpose of this study is to investigate long-term behavior of water quality parameters (liquid-phase DBPs and urea) and gas-phase $NCl_3$ after each process change. In addition, measurements are conducted before and after swimming team practice to examine the effects of a high bather loading on water quality.

## 2. Materials and Methods

### 2.1. Studied Pool Description

The swimming pool facility that was the focus of this study was located within a public recreational center. The pool had plan dimensions of 10-yards (9.15 m) wide and 25-yards (22.9 m) long. Water depth in the pool ranged between 3.5 and 5 feet (1.1–1.5 m). The volume of water in the pool was approximately 121,000 gallons (460 $m^3$). The pool was used by daily recreational swimmers, a group swimming course, swim team practice, and a water Zumba class. The swimming pool was managed by the pool staff, and no modifications were made to the heating, ventilation, and dehumidification system (HVAC). Calcium hypochlorite (Pulsar Plus Briquette, 68% available chlorine) was applied as the disinfectant/oxidant and sodium bisulfate (Clorox, 93.2%) was used for pH control. The studied pool facility relied on the local municipal water supply for filling and makeup water. The chemistry and composition of this public water supply, in terms of conventional constituents, is summarized in Table S1. The disinfectant (chlorine) feed rate was controlled by the oxidation–reduction potential (ORP). Figure S1 illustrates time-course measurements of ORP and residual chlorine across the duration of the study.

The secondary oxidant (Truox) applied in this study was a persulfate-based oxidizer (dipotassium peroxodisulphate) in the presence of an activator (potassium monopersulfate and a blend of cobalt and at least one of: copper, silver, iron, and manganese) that generates sulfate free radicals. The activator is a water-soluble metal-porphyrin that allowed application as a liquid. The feed rate of the oxidant was 2–5 pounds (0.9–2 kg) per 1,000,000 gallons (3800 $m^3$) of pool water per day. At the beginning, 2 gallons (8 L) of activator per 30,000 gallons (110 $m^3$) of pool water was introduced, then feed rate of the

activator was 8 oz (0.24 L) per hour. Both chemicals were added into pool with mixing tanks and chemical feed pumps. A simple schematic of the pool treatment process sequence and water sampling locations is presented in Figure S2.

### 2.2. Swimming Pool Water Sampling

Water samples were collected using 230 mL polyethylene terephthalate bottles with screw caps. No headspace was allowed because the studied compounds were volatile. Two kinds of water sample were collected every weekday morning. One was collected directly from the pool at the same spot every time (see Figures S2 and S3) and was referred to as 'pool water'; this sample was collected at the depth of 20 cm below water surface. The other kind of sample was collected downstream of treatment within the recirculation line; this sample was referred to as 'treated water.' Water samples were analyzed within 1 h of collection. Additional sampling included water samples that were collected from the pool prior to the evening swim team practice (5:00 p.m.) and the other was collected after practice (9:00 p.m.) on Monday and Thursday. The water sampling locations were selected to avoid interfering with pool operations and the activities of swimmers.

### 2.3. Air Sampling

An air quality monitoring device (NEMo, Ethera Labs, Crolles, France) was made available for measurements [22,23]. The NEMo device is able to accomplish measurements of gas-phase constituents using a proprietary SolGel nanoporous material on a sensor slide, which has the capacity to concentrate very low level (ppb) chemical compounds (600 $m^2$ of air exchange per gram material), then uses an electronic/optical system to measure the concentration of target compounds. For the measurements of gas-phase $NCl_3$ concentration, the sol–gel material was infused with potassium iodide (KI). The reaction between $NCl_3$ and KI results in formation of triiodide. A continuous increase in opacity in the sensor's chamber caused by triiodide was recorded in real time. The time-derivative of the opacity vs. time signal was used to calculate the 30 min moving average of the gas-phase $NCl_3$ concentration, which was reported every 10 min. The opacity signal was influenced by relative humidity (RH; opacity increases with RH), and the NEMo monitor was equipped with an algorithm to automatically correct the interference based on the signal from a built-in RH monitor. Based on manufacturer recommendations, only those measurements of gas-phase $NCl_3$ concentration collected under RH conditions between 30–60% were accepted and reported in this paper. The NEMo device was installed on a wall of studied pool facility. It was placed roughly 2.5 m above the swimming pool surface and 4 m from the swimming pool edge. Pool water sampling and NEMo locations relative to the swimming pool are illustrated in Figure S3.

In a previous study, the inclusion of KI in an impinger-based method for measuring gas-phase $NCl_3$ demonstrated that the signal was dominated by gas-phase $NCl_3$ and that there was minimal interference from other gas-phase oxidants, including other chloramines (inorganic and organic) and molecular oxygen [24]. No additional experiments were conducted in this study to examine the potential for interference by other gas-phase oxidants in the quantification of gas-phase $NCl_3$ by the NEMo device.

### 2.4. Water Sample Analysis

Membrane Introduction Mass Spectrometry (MIMS) was applied to quantify the concentration of volatile DBPs in pool water samples. The MIMS system used in this study comprised an Agilent GC-MS system (5975C mass-selective detector (MSD) and 6850 gas chromatograph (GC) (Agilent Technologies, Santa Clara, CA, USA) with a membrane injection device. The system was operated with electron ionization (70 eV). Compounds were identified using a mass spectrum scan mode (49 $\leq$ m/z $\leq$ 200) and further quantified with selected ion monitoring (SIM). The membrane injection device was installed in the GC; thus, the GC was used only for the temperature control of the membrane interface and

no chromatography was applied. Further details of the MIMS system that was used for analysis of volatile DBPs can be found in Shang et al. [25] and Weaver et al. [6].

Residual chlorine and DBP standards were prepared using an aqueous solution of free chlorine (Sigma-Aldrich, St. Louis, MO, U.S., 5% sodium hypochlorite (NaOCl)). Free chlorine was diluted to 500 mg/L as $Cl_2$ as a stock solution with deionized and nanopure water and covered with aluminum foil in a glass-stoppered bottle, then stored at 4 °C in a refrigerator. Standard chlorine solutions were then produced by titrimetric methods every week [26]. Standard solutions of monochloramine ($NH_2Cl$) and $NCl_3$ were freshly prepared follow the method described in Shang and Blatchley III [25]. Standard solutions of CNCl were produced by 30 min chlorination (free chlorine, 5 mg/L as $Cl_2$) of a glycine solution (Sigma-Aldrich, St. Louis, MO, USA, 0.5 mg/L as N) at pH 7; pH was controlled using a phosphate buffer [27]. Standard curves were created for the MIMS system for each of the studied compounds at their unique m/z peak, thus allowing the calculation of the concentration from the slope of the standard curve and the reported abundance signal for each analyte in a sample. The detection limits for volatile DBPs measured by MIMS were defined on the basis of a 2:1 signal-to-noise ratio. The detection limits were 60, 3.6, and 0.8 μg/L (as $Cl_2$) for $NH_2Cl$, $NCl_3$, and CNCl, respectively. Urea measurements were conducted with water samples followed by the digestion procedure involving the antipyrine and colorimetric analysis described by Prescott and Jones [28].

## 3. Results and Discussion

### 3.1. Liquid-Phase $NH_2Cl$ and $NCl_3$

The term 'combined chlorine' is commonly understood to refer to inorganic chloramines: $NH_2Cl$, dichloramine ($NHCl_2$), and $NCl_3$. Inorganic chloramines in swimming pool water are mainly the products of reactions between free chlorine and organic-N compounds [25]. Previous research has revealed typical concentrations of $NH_2Cl$ and $NCl_3$ found in public swimming pools from below detection limit to 1880 μg/L as $Cl_2$ and below detection limit to 377 μg/L as $Cl_2$, respectively [6].

Liquid-phase $NH_2Cl$ and $NCl_3$ were consistently measurable throughout the study period. The results of time-course monitoring of the liquid-phase $NH_2Cl$ and $NCl_3$ are illustrated in Figure 1. The concentration of $NH_2Cl$ gradually increased after the replacement of filter media. In general, swimmer numbers did not correlate strongly with the measured concentrations of $NH_2Cl$. Significant reductions in $NH_2Cl$ concentration were observed after the introduction of the activator. After replacing the sand filter medium with activated filter media, a slight improvement ins $NCl_3$ concentration was observed for a period of about two weeks, but the $NCl_3$ concentration increased afterward. After the addition of the secondary oxidizer, the liquid-phase concentration of $NCl_3$ diminished and remained stable for four weeks. This suggests that the rates of formation and decay of $NCl_3$ had approached similar values, thereby resulting in a steady-state condition during this period. A further reduction in $NCl_3$ concentration was detected following the application of the activator. After the feeding of the secondary oxidizer was discontinued, the concentration of $NCl_3$ in 'pool water' samples gradually increased. This suggests that the secondary oxidizer contributed to the degradation of $NCl_3$, its precursors, or both. Additionally, it appears that the secondary oxidizer did not remain in the water to participate in the oxidation of $NCl_3$ or its precursors, suggesting that there was an active demand for the secondary oxidizer and its activator. There was no clear correlation between swimmer count and the concentrations of $NH_2Cl$ and $NCl_3$ in the pool water.

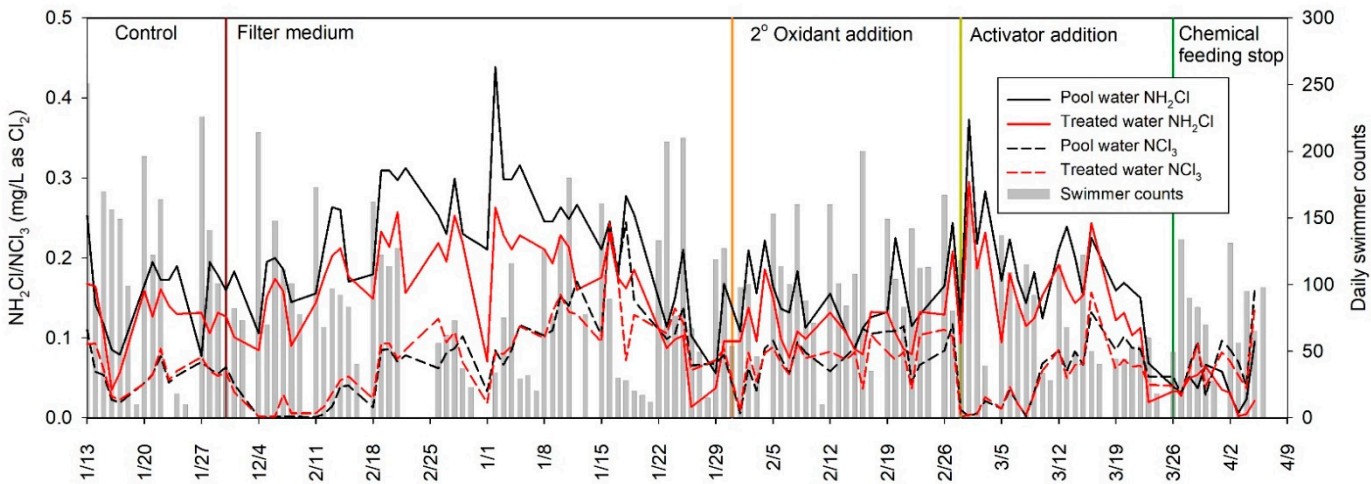

**Figure 1.** Time-course monitoring of the liquid-phase NH$_2$Cl and NCl$_3$ concentration in the studied pool, as measured by MIMS. The black solid line represents measurements of NH$_2$Cl concentration from pool water. The red solid line represents measurements of NH$_2$Cl concentration from treated water. The black dashed line represents measurements of NCl$_3$ concentration from pool water. The red dashed line represents measurements of NCl$_3$ concentration from treated water. Gray vertical bars represent hourly swimmer counts.

The measurements of liquid-phase NCl$_3$ before swimming practice and after swimming practice are illustrated in Figure 2. Concentrations of the liquid-phase NCl$_3$ after swimming practice were generally higher than before practice, suggesting a swimmer impact on NCl$_3$ in pool water. The difference of NCl$_3$ concentrations between before and after practice became minor after the introduction of the activator, suggesting that it might reduce the swimmer impact on NCl$_3$ concentrations in the pool.

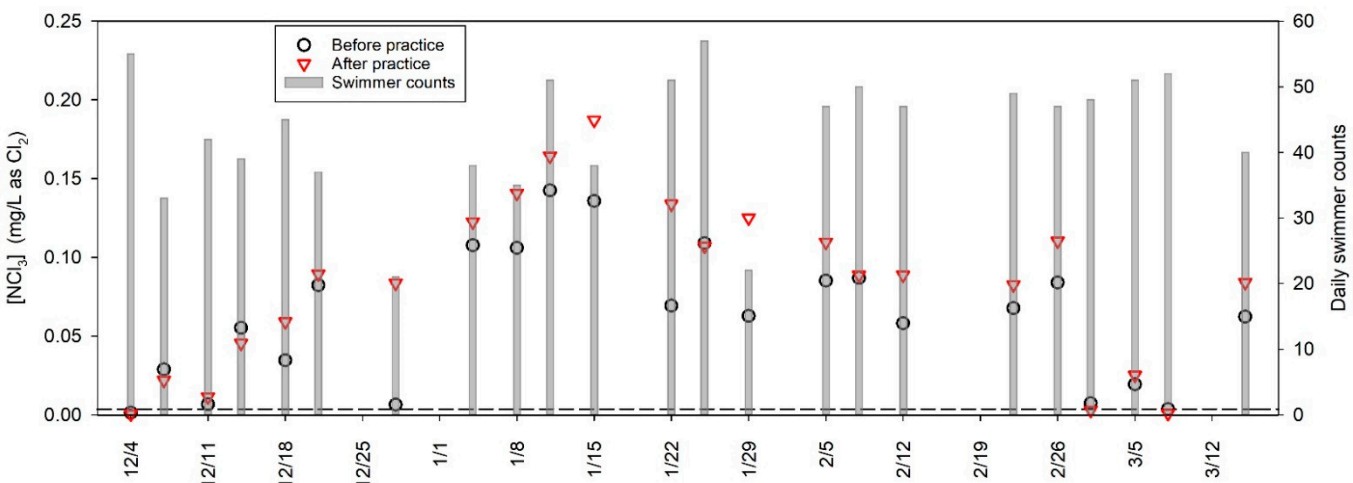

**Figure 2.** Time-course monitoring of the liquid-phase NCl$_3$ concentration before and after swimming team practice sessions by MIMS. Empty circle symbols represent measurements of NCl$_3$ concentration before swimming practices. Empty triangle symbols represent measurements of NCl$_3$ concentration after swimming practices. Gray vertical bars represent swimmer counts.

### 3.2. Liquid-Phase CNCl

The results of time-course monitoring of CNCl concentration in pool water are shown in Figure 3. The CNCl concentrations in 'treated water' were relatively consistent and low during the study period. However, a substantial variation and higher concentrations of CNCl were noticeable in 'pool water' samples. It is hypothesized that the oxidation of CNCl

occurred during the time spent by the water in the recirculation line where the residual free chlorine concentration would be relatively high; however, CNCl reformation occurred in the pool. A substantial reduction in CNCl level in the studied pool was observed after the addition of secondary oxidizers, and the measurements of the liquid-phase CNCl in the 'pool water' and 'treated water' samples were almost equal after that time.

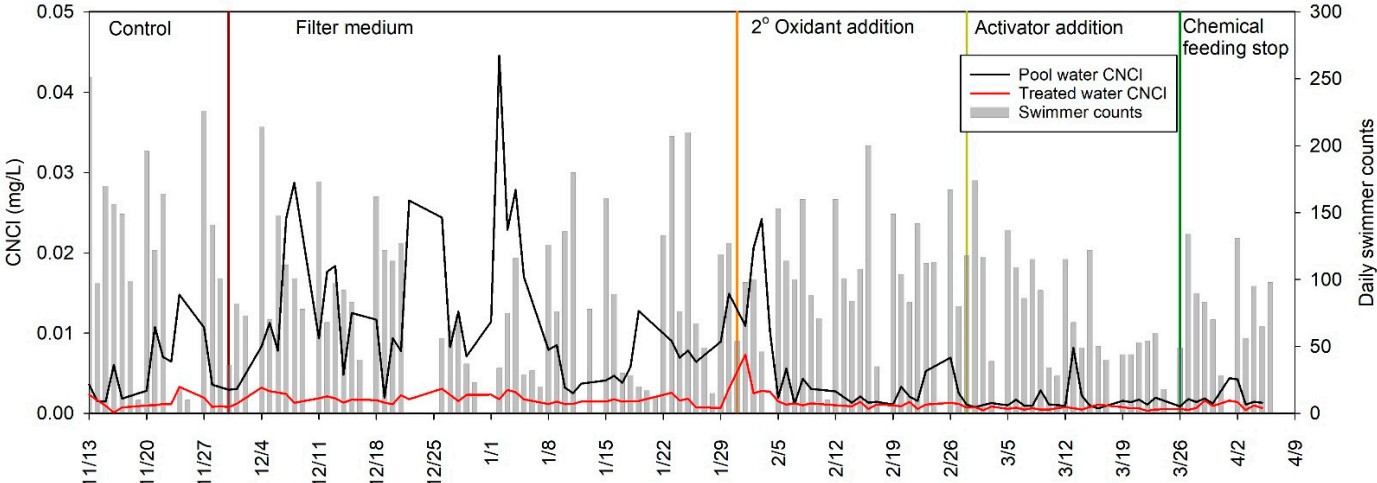

**Figure 3.** Time-course monitoring of the liquid-phase CNCl concentration at the studied pool measured by MIMS. The black solid line represents measurements of CNCl concentration from pool water. The red solid line represents the measurements of CNCl from treated water. Gray vertical bars represent swimmer counts.

Previous studies have shown that CNCl and free chlorine in water samples tend to be negatively correlated [19,29]; that is to say, maintaining high free residual chlorine can be used to limit the concentration of CNCl because free chlorine plays a critical role in the decay of CNCl. Specifically, hypochlorite ion ($OCl^-$) displays the ability to catalyze the oxidation of CNCl to cyanate [29]. However, it is important to recognize that chlorine is also critical for the formation of CNCl from reactions with amino acids and uric acid [3,30]. When temperature = 25 °C, free chlorine concentration = 0.5 mg/L (as $Cl_2$), and pH = 7, the half-life of CNCl in water has been reported to be roughly 60 min [29]. For perspective, Weaver et al. [6] measured the liquid-phase CNCl concentrations in water samples collected from a range of public pool waters and the reported concentrations ranged from below detection limit to 194 μg/L.

### 3.3. Urea

Urea is the dominant organic-N compound found in human urine and sweat, especially by mass. Urea is introduced to pool water by the excretion of human body fluids, especially urine and sweat. Urea is also found in skin tissues as a component of 'natural moisturizing factor' that is used by the body to maintain skin moisture [31,32]. Thus, contact between swimmer skin and pool water may also represent a significant source of urea introduction to pools. A substantial fraction of $NCl_3$ in swimming pool water has been shown to be attributable to the reactions of free chlorine and urea [33,34]. De Laat et al. [35] reported the urea concentration in swimming pools to be in the range 0.12–3.6 mg/L. Weng and Blatchley III [36] reported urea concentrations that ranged from 0.07–0.15 mg/L in an indoor swimming pool under conditions of heavy use. It is known that urea concentrations are associated with the bather load in the swimming pool, as well as water replacement practices, and the hygiene practices of swimmers. Large numbers of swimmers and slow water replacement rates generally correspond with a high urea concentration [36].

The results of time-course monitoring of urea concentrations are summarized in Figure 4. Overall, urea concentrations in 'treated water' samples were slightly lower than

pool water samples; the reasons for this behavior are not entirely clear. A regular, weekly pattern emerged wherein the urea concentration was generally lowest on Monday and gradually increased through the remainder of the week. Since urea is slow to react with chlorine, this behavior suggested that urea slowly accumulated during periods of heavy pool use (weekdays), it but degraded over the weekend when the pool was used only sparingly. The fact that the concentration of urea in pool water samples collected after practice was consistently higher than the concentration in samples collected at the same location immediately before practice supports this hypothesis.

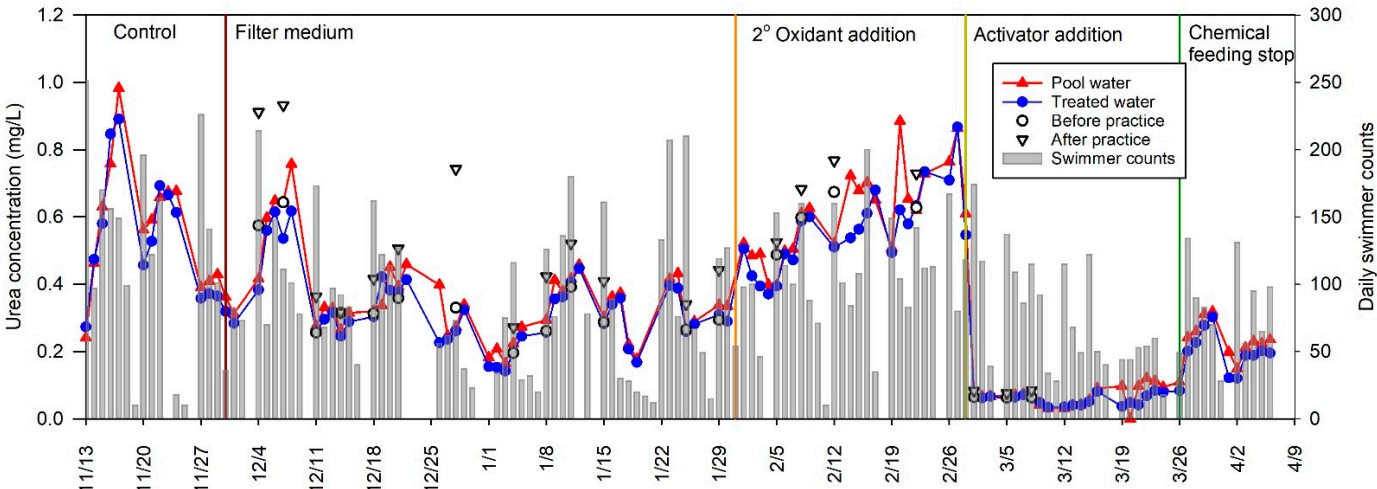

**Figure 4.** Time-course monitoring of urea concentrations. Red triangle symbols represent measurements of urea concentration in pool water samples. Blue circle symbols represent measurements of urea concentration in treated water samples. Empty circle symbols represent measurements of urea concentration in pool water samples collected before swimming practices. Empty triangle symbols represent measurements of urea concentration in pool water samples collected after swimming practices. Gray vertical bars represent swimmer counts.

Urea concentration decreased after the filter media was changed, as shown in Figure 4. The lowest concentrations observed during this stage were found during the Christmas and New Year holiday periods, when relatively few swimmers used the pool. Urea concentration was not diminished after the addition of secondary oxidizers, but effective urea reductions were observed after the use of activator, when urea concentration was reduced to as low as 0.05 mg/L. A steady, immediate increase in urea concentration was observed after the feeding of the secondary oxidizer and the activator was stopped. This implies that the secondary oxidizer and activator were consumed and that the control of urea concentrations within the pool depends on the continuous addition of these chemicals.

### 3.4. Gas-Phase $NCl_3$

Time-course monitoring of gas-phase $NCl_3$ concentration measured by NEMo is illustrated in Figure 5. The hourly counts of the number of swimmers in the pool during the studied period were also display in Figure 5. To reiterate, the manufacturer of the air quality monitor indicated that the device only provides accurate measurements of gas-phase $NCl_3$ concentrations when relative humidity (RH) is within 30–60%. When RH is higher than 60%, the measurement of gas-phase $NCl_3$ is overestimated. On the other hand, when RH is lower than 30%, the reading of gas-phase $NCl_3$ is underestimated. The HVAC system in the pool facility was unable to maintain consistent humidity for an extended period. Humidity was often out of the 30–60% range and unstable after 12/10. The design of the HVAC system in the studied pool facility involved the use of outdoor air to help to control the RH in the pool facility. Rapid changes of humidity are often observed in the upper Midwest, especially during the cold winter months. The RH constraint of the NEMo device

limited access to reliable measurements of gas-phase $NCl_3$ during the study period to only those times where the RH condition was satisfied (see Figure S4). Thus, only measurements of gas-phase $NCl_3$ concentration corresponding to times where RH was within 30–60% are presented in Figure 5.

The data presented in Figure 5 illustrate two relevant phenomena. First, high gas-phase $NCl_3$ concentrations were generally observed during the period of high swimmer numbers and immediately thereafter. The activity of swimmers in a pool is believed to promote the transfer of volatile compounds from the liquid phase to the gas phase, largely as a result of the mechanical mixing imparted by the swimmers on the water near the air–water interface. When large numbers of swimmers were present in the pool, $NCl_3$ transfer from water to air was enhanced. The surge of gas-phase $NCl_3$ concentration was most evident after periods when more than 40 swimmers were in the pool; gas-phase $NCl_3$ concentrations during these times were as high as 1.8 mg/m$^3$. For perspective, an upper limit for gas-phase $NCl_3$ concentration of 0.5 mg/m$^3$ was recommended by the World Health Organization (WHO) [37], and 0.3 mg/m$^3$ was suggested by Bernard et al. [38]. The high concentrations of gas-phase $NCl_3$ observed in the studied pool could potentially cause adverse health effects for swimmers, lifeguards, and spectators. Secondly, changes in the water treatment processes had little or no effect on gas-phase $NCl_3$ concentration. The peak concentrations during each period did not appear to decrease from Figure 5a–d. Two factors are believed to have contributed to this observation. First, the liquid-phase $NCl_3$ concentration increased during the periods of heavy use (see Figure 2). Second, the population of swimmers changed during the study period, such that in the latter stages of the study, there was increased use on the part of a high school swimming team. It is assumed that type of swimmer and type of swimmer activity influences the transfer of $NCl_3$ from water to air in pool facilities. In this study, only the number of swimmers were recorded, not the type or age of swimmer. Thus, the level of contribution of liquid-to-gas transfer by different types of swimmers could not be examined.

The values of gas-phase $NCl_3$ reported in this paper were consistent with previous reports. For example, Zare Afifi and Blatchley [19] reported concentrations of gas-phase $NCl_3$ from undetectable to 0.62 mg/m$^3$, with the mean concentration of 0.15 mg/m$^3$ in a high school swimming pool. Seys et al. [39] measured gas-phase $NCl_3$ concentrations ranging from 0.2 to 1.4 mg/m$^3$ in a local swimming pool of Leopoldsburg, Belgium. Lévesque et al. [17] also measured gas-phase $NCl_3$ at different time periods at a pool in Québec, Canada; the mean concentrations in the morning, afternoon, and night were 0.35, 0.36, and 0.51 mg/m$^3$, respectively. Fornander et al. [40] reported gas-phase $NCl_3$ concentrations ranging from 0.04 to 0.36 mg/m$^3$ in nine pool facilities; the mean concentration was 0.2 mg/m$^3$. Nordberg et al. [41] also investigated human exposure to gas-phase $NCl_3$ in two groups of people (recreational swimmers and pool workers). At the time of their exposure, mean gas-phase $NCl_3$ concentration for non-regular swimmers was 0.23 mg/m$^3$ and mean gas-phase $NCl_3$ concentration for pool workers was 0.19 mg/m$^3$. Reported gas-phase concentrations were related to the liquid-phase free chlorine concentration, numbers of bathers, and HVAC system characteristics [1].

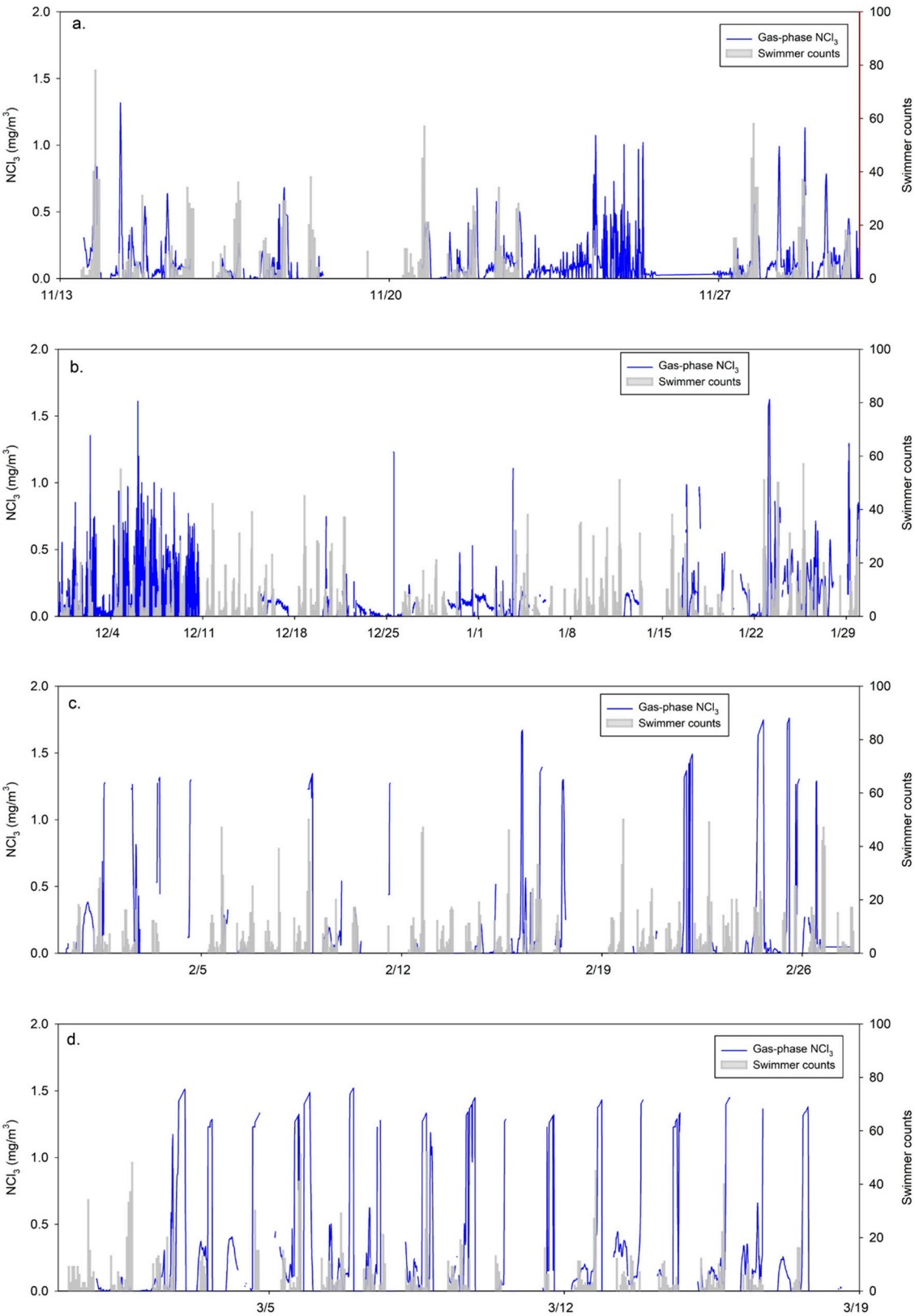

**Figure 5.** Time-course monitoring of gas-phase $NCl_3$ concentration: (**a**) control period, (**b**) filter media replaced period, (**c**) secondary oxidant addition period, and (**d**) activator addition period. Blue lines represent measurements of gas-phase $NCl_3$ concentration collected by NEMo. Gray vertical bars represent swimmer counts.

## 4. Conclusions

After the application of secondary oxidizers, reductions in free chlorine and total chlorine in the swimming pool water were observed. The concentration of liquid-phase CNCl was reduced substantially after the addition of the oxidizers. The concentrations of $NCl_3$ in the water samples also declined after the introduction of the secondary oxidizers. The lower liquid-phase concentrations of these compounds apparently depended on the presence of the secondary oxidant, as increases in the concentrations of these DBPs were seen once the secondary oxidant feed was discontinued. The addition of the activator also resulted in small reductions in the concentrations of $NCl_3$ and CNCl. The introduction of the activator in the pool substantially diminished the concentration of urea. Secondary oxidizers coupled with the activator could be an effective treatment process to limit the accumulation of DBPs and their precursors in swimming pools.

There was clear evidence that swimmer activity affects the concentration of gas-phase $NCl_3$. Specifically, the gas-phase $NCl_3$ concentration was observed to increase as a result of increased swimmer number. This was probably attributable to increases in the liquid-phase $NCl_3$ concentration as well as the effects of mechanical mixing imparted by swimmers on water near the air–water interface. However, changes in water treatment processes did not improve the air quality in the studied pool. In all, NEMo could be a reliable air quality monitor when relative humidity is within 30% to 60%. IAQ in indoor pool facilities is strongly associated with gas-phase $NCl_3$ concentration; the ability to measure gas-phase $NCl_3$ in near real time offers the potential to improve IAQ in indoor pool facilities.

**Supplementary Materials:** The following supporting information can be downloaded at: https://www.mdpi.com/article/10.3390/w14030335/s1. Table S1: Source water quality analysis; Figure S1: Time-course monitoring of free chlorine, total chlorine and ORP at the studied pool water; Figure S2: Schematic of water treatment processes with chemical dosing points and locations of pool water sampling locations; Figure S3: Water sampling and NEMo locations in the studied pool facility; Figure S4: Time-course monitoring of relative humidity (RH) at the studied pool facility.

**Author Contributions:** Conceptualization, L.T.L. and E.R.B.III; methodology, L.T.L. and E.R.B.III; validation, L.T.L. and E.R.B.III; formal analysis, L.T.L. and E.R.B.III; investigation, L.T.L. and E.R.B.III; resources, E.R.B.III; data curation, writing—original draft preparation, L.T.L.; writing—review and editing, L.T.L. and E.R.B.III; visualization, L.T.L. and E.R.B.III; supervision, E.R.B.III; project administration, E.R.B.III; funding acquisition, E.R.B.III. All authors have read and agreed to the published version of the manuscript.

**Funding:** This research was funded by PoolPak through a testing agreement, number 18024503.

**Acknowledgments:** The authors are grateful to the Ethera labs, PoolPak, and Spear Corporation for their technical support and guidance.

**Conflicts of Interest:** The authors declare no conflict of interest.

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
