# Peer review of "Long-Term Monitoring of Water and Air Quality at an Indoor Pool Facility during Modifications of Water Treatment"

_water, doi:10.3390/w14030335_

Round 1
Reviewer 1 Report
In the MS, the authors conducted an experiment to monitor chemical water and air quality in an indoor swimming pool that was undergoing changes in water treatment. MS is interesting and also applicable to the general public, but also has some points that MS could do better. Here are some comments.
What happens to the microbial water quality of the pool water when you replace or upgrade one or more components of the water treatment system?
Where is the fixed location for pool sampling? How did you choose this particular, unique location?
Is the water sampling location in the same location as the NEMo?
Have you analyzed the microbial indicators of water quality?
Your sample is approximately 230 ml in size. What is the protocol for sampling volume in the 460 m3 pool? Do you do replicates of the samples?
What is the air quality at a relative humidity of < 30% and > 60%?
Reviewer 2 Report
Please describe terms like ORP and IAQ.
Manufactures and concentrations of materials in the pool should be added.
Description of standards ( Brand and concentration) and method used for analysis needs to be expanded with your method detection limit and error analysis ( i.e. were samples for the pool run in triplicate?) .
Was any work done to show that the KI method was specific to NCl3 and not affected by other possible chlorine containing compounds yielding false higher readings.
The air sampling work was not for the duration of the study and added little to the paper and possibly should be redone or removed from the paper as written. If redone , it may make for a useful stand alone article.
Reviewer 3 Report
Manuscript titled as "Long-Term Monitoring of Water and Air Quality at an Indoor Pool Facility During Modifications of Water Treatment" deals with very important topic – relation of the water treatment procedure and quality of water and air parameters in the indoor swimming pool.
The manuscript is overall well written and structured and biggest part of the topics is well described. However, I think there is still a lot of space for improvements:
My suggestions/comments:
- Figure S1 – unit of ORP is missing
- general water quality parameters of the local public water supply (e.g. was it hard or soft water, source pH, TOC/DOC, SUVA etc…) are missing
- Authors wrote, Line 85:” Disinfectant (chlorine) feed rate was controlled by ORP (see Figure S1).” However, on the Figure S1 authors did not mention the chlorine feed rate (or usual chlorine dose in the pool) and also did not explain how the chlorine rate was controlled. Please, explain it and give us the data bout the Cl dose!
- positions of the dosing place in the water treatment scheme is not clear. It would be of great assistance if authors draw a simple schematic view of the pool and basic treatment steps sequence within the dosing place of all the chemicals would be marked. Also, the fresh water supply rate dynamics should be mentioned – is it continuous 24/7/30 or is it changing during the day…
- more details (exact substances and doses, e.g. mg/L; statement “a persulfate-based oxidizer is not enough!)) are necessary in the lines 86-88: ”The secondary oxidant applied in this study is a persulfate-based oxidizer in the presence of an activator that generates sulphate free radicals. The activator is a water-soluble Metal-Porphyrin that allowed the oxidizer to be applied as liquid form”
- Figure 1 – parameters in the description were not well described and also the discussion of the results is rather poor – the authors should discuss these results in more detail and answer the question of why such changes occur
- explanation of the situation when the dosing of secondary oxidizer was stopped is missing – what is the purpose of this examination? Is the dosing of the activator also stopped in that time? Why are results different then in case of the first weeks without oxidizer dosing?
- Figure 5 - explanation is needed – why parameters were monitored only in the period 11/13 to 12/4? It should be very interesting to see what is happening during oxidizer and activator dosing…
- in Conclusions – line 276/277 there is a statement: ”After application of secondary oxidizers, reductions of free chlorine and total chlorine in the swimming pool water were observed”. Authors did not show us any proof for this statement!
Reviewer 4 Report
On account of the manuscript WATER-1534488, entitled “Long-Term Monitoring of Water and Air Quality at an Indoor Pool Facility During Modifications of Water Treatment” by Lester T. Lee et al., the authors investigated long-term behavior of water quality parameters (liquid-phase volatile disinfection byproducts (DBPs) and urea) and air quality parameters as gas-phase NCl3 after each process change in swimming pool. The topic is important to conduct the environmental management of swimming pool facility, and human health risk management. The manuscript was well written and designed, and the authors got interesting results. After careful consideration, I made a decision that the manuscript is acceptable for publication in its present form.
Special remarks:
‧ The present manuscript evaluated the changes in water and air quality associated with changes in water treatment at an indoor chlorinated swimming pool facility.
‧ The authors hypothesis, water and air quality in a swimming pool facility can be improved by renewing or enhancing one or more components of water treatment, is considered to new view point and interesting.
‧ The present manuscript provided useful prospects to better understandings for the environmental management of swimming pool conditions and human health risk asessment.
‧ The interpretation of the evidence and arguments presented and conclusions are sufficient.
‧ The references cited relevant and up to date.
‧ The tables and/or figures are useful, necessary, and good quality.
Round 2
Reviewer 2 Report
Needed corrections and additions have been made. Thank you!
Reviewer 3 Report
The authors have fulfilled almost all the remarks I had, and looking at the remarks and answers of other reviewers, I can confirm that this article is now of better quality than in the previous version, and I support its publication.